# Ethylene Differentially Modulates Hypoxia Responses and Tolerance across *Solanum* Species

**DOI:** 10.3390/plants9081022

**Published:** 2020-08-13

**Authors:** Sjon Hartman, Nienke van Dongen, Dominique M.H.J. Renneberg, Rob A.M. Welschen-Evertman, Johanna Kociemba, Rashmi Sasidharan, Laurentius A.C.J. Voesenek

**Affiliations:** Plant Ecophysiology, Institute of Environmental Biology, Utrecht University, Padualaan 8, 3584 CH Utrecht, The Netherlands; n.s.vandongen@students.uu.nl (N.v.D.); d.m.h.j.renneberg@uu.nl (D.M.H.J.R.); R.A.M.Welschen@uu.nl (R.A.M.W.-E.); johanna.kociemba@crick.ac.uk (J.K.)

**Keywords:** ethylene, flooding, hypoxia, phytoglobin, VII Ethylene Response Factor, PRT6 N-degron pathway of proteolysis, *Solanum tuberosum*, *Solanum lycopersicum*, *Solanum dulcamara*

## Abstract

The increasing occurrence of floods hinders agricultural crop production and threatens global food security. The majority of vegetable crops are highly sensitive to flooding and it is unclear how these plants use flooding signals to acclimate to impending oxygen deprivation (hypoxia). Previous research has shown that the early flooding signal ethylene augments hypoxia responses and improves survival in Arabidopsis. To unravel how cultivated and wild *Solanum* species integrate ethylene signaling to control subsequent hypoxia acclimation, we studied the transcript levels of a selection of marker genes, whose upregulation is indicative of ethylene-mediated hypoxia acclimation in Arabidopsis. Our results suggest that ethylene-mediated hypoxia acclimation is conserved in both shoots and roots of the wild *Solanum* species bittersweet (*Solanum dulcamara*) and a waterlogging-tolerant potato (*Solanum tuberosum*) cultivar. However, ethylene did not enhance the transcriptional hypoxia response in roots of a waterlogging-sensitive potato cultivar, suggesting that waterlogging tolerance in potato could depend on ethylene-controlled hypoxia responses in the roots. Finally, we show that ethylene rarely enhances hypoxia-adaptive genes and does not improve hypoxia survival in tomato (*Solanum lycopersicum*). We conclude that analyzing genes indicative of ethylene-mediated hypoxia acclimation is a promising approach to identifying key signaling cascades that confer flooding tolerance in crops.

## 1. Introduction

Global food demand is expected to double by 2050, as a result of the rising world population and shifting human diets [1]. However, future food security is highly challenged by a decline in arable land and an increase in crop losses, due to the consequences of a changing climate [1,2]. Indeed, elevated temperatures and extreme precipitation patterns due to climate change have led to an increase in the frequency and severity of flooding events [3]. Floods strongly impair agricultural crop production and exacerbate the food security crisis [3,4]. In order to minimize crop losses and safeguard global food security, it is therefore paramount to develop flood-tolerant crops in the near future [5]. Flooding tolerance in plants is regulated by variety of signals that mediate responses to combat oxygen (O_2_) deprivation (hypoxia) and ameliorate toxic reactive oxygen species (ROS) [4,6]. Submerged terrestrial plants ultimately encounter hypoxia, due to impaired gas diffusion under water, and flooding survival therefore strongly depends on traits that enhance hypoxia tolerance [4,7]. In addition, passive ethylene entrapment acts as a rapid signal for submergence, and regulates many flood adaptive responses that include morphological and anatomical modifications to prevent hypoxia [8]. Ethylene also mediates metabolic hypoxia acclimation in the wetland species *Rumex palustris* and the model species Arabidopsis (*Arabidopsis thaliana*) [9,10,11]. In Arabidopsis, ethylene-mediated hypoxia tolerance is conserved in root and shoot meristems and is associated with enhanced expression of a conserved set of hypoxia-responsive genes when O_2_ levels decline [10,12]. These hypoxia adaptive genes are predominantly controlled by the group VII Ethylene Response Factor (ERFVII) transcription factors RELATED TO APETALA2.2 (RAP2.2), RAP2.12 and RAP2.3 [13,14]. ERFVIIs are typically known to control a variety of developmental growth [15,16,17], biotic [18,19] and abiotic stress responses in plants [20,21,22,23]. Arabidopsis ERFVIIs possess a conserved N-terminal sequence that regulates their turnover through the PROTEOLYSIS6 (PRT6) N-degron pathway under O_2_ and nitric oxide (NO) replete conditions [24,25,26]. When the cellular levels of either O_2_ or NO decline, ERFVII proteolysis is restricted and these proteins stabilize, ultimately activating target hypoxia adaptive gene expression [24,25,26]. Ethylene can already stabilize ERFVIIs prior to hypoxia through active NO reduction, by enhancing levels of the NO-scavenging protein PHYTOGLOBIN1 (PGB1) [10]. In addition, ethylene can promote ERFVII production through transcriptional activation. Collectively, these results showed that the cascade initiated by ethylene upon submergence is crucial to prevent N-degron targeted ERFVII proteolysis by the increased production of the NO-scavenger PGB1. This ethylene-induced ERFVII accumulation, in turn, augments the transcription of downstream target genes to improve energy homeostasis and oxidative stress tolerance during hypoxia and re-oxygenation in Arabidopsis [10,27,28,29].

Interestingly, this work in Arabidopsis revealed that if ethylene was not able to enhance PGB1 or ERFVII levels (through mutant analysis), ethylene was also unable to augment the transcriptional hypoxia response and promote hypoxia tolerance [10]. Similarly, the inability to enhance hypoxia tolerance through early ethylene signaling correlated with the absence of ethylene-induced expression of *PGB1*, *RAP2.12* and hypoxia adaptive gene orthologues in the terrestrial wild plant species *Rumex acetosa* [9]. Contrastingly, but similar to Arabidopsis, *R. palustris* uses the ethylene signal to promote hypoxia survival and this corresponded with *PGB1*, *RAP2.12* and hypoxia adaptive transcript induction [9]. Finally, natural variation for higher flooding tolerance is frequently found to be correlated with enhanced ethylene signaling [30,31], PGB1 levels and NO removal [32,33], ERFVII levels [31,34] and hypoxia adaptive gene transcripts [30,35,36,37,38] in crop and ornamental plant species. Taken together, the mechanism of ethylene-mediated hypoxia tolerance could be instrumental in enhancing flooding tolerance of intolerant crops through the manipulation of ethylene responsiveness of *PGB1* and *ERFVII* genes and its downstream signaling targets.

Potato (*Solanum tuberosum*) and tomato (*Solanum lycopersicum*) are the most economically important cultivated vegetable crops in the world (e.g., the global tomato export value exceeded USD 9 billion in 2019 [39]), but are both considered to be highly intolerant to abiotic stresses, including flooding [40,41]. Conversely, the closely related wild plant species bittersweet (*Solanum dulcamara*) is highly tolerant to a variety of abiotic stresses [42]. All three species display multiple (ethylene-mediated) flood adaptive responses [41,43,44], but still show strong differences in flooding tolerance. However, whether ethylene also contributes to metabolic hypoxia acclimation and if this correlates with flooding tolerance in these *Solanum* species is currently unknown. Several studies showed that genes involved in ethylene signaling and ethanolic fermentation are induced in response to flooding and hypoxia, but not consistently across all plant tissues in these *Solanum* species [37,40,41,45,46].

Here, we investigated whether these cultivated and wild *Solanum* species integrate the early flooding signal ethylene to control subsequent hypoxia acclimation. To do so, we studied transcript levels of a selection of representative marker gene orthologues that control ethylene-mediated hypoxia responses in Arabidopsis. In addition, we tested whether an ethylene pre-treatment was able to enhance survival during subsequent hypoxia in roots and shoots of several *Solanum* species. Interestingly, our results revealed that ethylene contributes to hypoxia acclimation responses in bittersweet and a waterlogging-tolerant potato cultivar, but does not in tomato and the roots of waterlogging-sensitive potato cultivar. We propose that the mechanism of ethylene-mediated hypoxia acclimation could be instrumental to uncover signaling cascades that confer flooding tolerance in crops.

## 2. Results

### 2.1. Elite Potato Cultivars Show Variation in Waterlogging Tolerance

To unravel whether ethylene-mediated hypoxia acclimation contributes to potential flooding tolerance differences in potato, we first explored whether variation in flooding tolerance exists across six elite potato cultivars. Potato plants grown from tubers were waterlogged for up to 12 days and shoot growth parameters were scored every 3 days. Overall, the cultivars Festien and Avarna retained the highest shoot biomass and leaf area, whereas the cultivars Seresta and Ambition showed the highest loss in shoot performance during waterlogging (Figure 1). The differences between the tolerant cultivar Festien and sensitive Seresta were the most consistent for both biomass and leaf area at every recorded time point during the full duration of the waterlogging experiment. Since ethylene typically accumulates in flooded plant tissues [8], we explored whether differences in ethylene-mediated hypoxia responses could contribute to this contrasting waterlogging tolerance among potato cultivars. For this, we selected the tolerant Festien and sensitive Seresta for further transcriptional analysis.

### 2.2. Ethylene Differentially Mediates the Transcriptional Hypoxia Response in Potato Roots

In Arabidopsis, augmentation of the transcriptional hypoxia response by ethylene depends on increased levels of ERFVIIs and PGB1 prior to hypoxia. Here, we selected several potato orthologues of Arabidopsis genes associated with this ethylene-controlled hypoxia tolerance as markers for the integration of ethylene signaling with the transcriptional hypoxia response. These included: *ETHYLENE RESPONSE 2* (*ETR2*) [47], to test whether ethylene indeed led to ethylene-dependent signaling; *PGB1* as a marker gene for enhanced NO-scavenging capacity [48]; *RAP2.12* to test whether ethylene induces potato ERFVIIs at the transcriptional level and *PYRUVATE DECARBOXYLASE 1* (*PDC1*), encoding the rate limiting protein for hypoxia-induced fermentation [49], as a representative marker gene for the transcriptional hypoxia response. Since *ETR2* and *PGB1* are also among the ERFVII-controlled hypoxia-responsive genes in Arabidopsis [13], they could additionally be interpreted as markers for the hypoxia response and function as an indication for enhanced ERFVII levels, prior to hypoxia.

To unravel how ethylene modulates hypoxia responses in potato, plants were pre-treated with air or ethylene followed by 4 h of hypoxia. Ethylene enhanced the transcript abundance of *ETR2* and *RAP2.12* orthologues in shoot and root tissues of both potato cultivars (Figure 2), showing that the ethylene treatment quickly leads to ethylene-dependent signaling. Interestingly, *PGB1* transcripts were enhanced in response to ethylene in both tissues of the more tolerant cultivar Festien, but were not enriched in the roots of the sensitive cultivar Seresta (Figure 2). Moreover, ethylene pre-treatment also enhanced the hypoxia response of *ETR2*, *PGB1* and *PDC1* transcripts in both the shoots and roots of the tolerant cultivar Festien. However, while ethylene pre-treatment did enhance the hypoxia response of *PGB1* and *PDC1* in the shoots of sensitive cultivar Seresta (Figure 2a), ethylene had no beneficial effect on the hypoxia-responsiveness of *ETR2*, *PGB1* and *PDC1* transcripts in the root tissues (Figure 2b). Together, these data suggest that the capacity to enhance root hypoxia response genes through ethylene signaling corresponds to higher waterlogging tolerance in these two potato cultivars, and could depend on *PGB1* induction to modulate enhanced metabolic hypoxia acclimation.

### 2.3. Ethylene Differentially Modulates Hypoxia Survival in Solanum Seedlings

While an ethylene-enhanced transcriptional hypoxia response coincides with increased waterlogging tolerance in the previously described potato cultivars, it is unclear whether it also corresponds to ethylene-mediated hypoxia tolerance in *Solanum* species. Previous work has shown that early ethylene signaling is essential to improve the hypoxia tolerance of Arabidopsis and bittersweet roots, but not tomato roots [10]. Here, we tested the root hypoxia tolerance of additional *Solanum* species and the shoot hypoxia tolerance of both bittersweet and tomato. Ethylene pre-treatment strongly enhanced root tip survival in both bittersweet and eggplant (*Solanum melongena*) seedlings, but had no effect on tomato (Moneymaker) and *Solanum pennellii* root tip survival (Figure 3a,b). Furthermore, the ethylene pre-treatment improved the survival and biomass retention (fresh weight) of bittersweet shoot tissues during hypoxia (Figure 3c–e). Similar to root tips, ethylene had no effect on hypoxia tolerance of tomato shoots (Figure 3f,g). Collectively, these results indicate that the ethylene-mediated hypoxia tolerance is conserved in both the shoots and roots of the flood tolerant plant bittersweet, but not in the tomato cultivar Moneymaker.

### 2.4. Ethylene Differentially Mediates the Transcriptional Hypoxia Response between Bittersweet and Tomato

Next, we assessed whether the contrasting tolerance between tomato and bittersweet corresponds to how these species use an early ethylene signal to control its subsequent transcriptional hypoxia response. Similar to the analysis in potato, we quantified transcript levels of the representative marker genes studied earlier (Figure 2), but now also included the orthologues of ethylene response marker *ACC OXIDASE 1* (*ACO1*), NO-scavenging genes *PGB2* and *PGB3* [50], the ERFVII *RAP2.3* and the hypoxia markers *SIMILAR TO RCD ONE 5* (*SRO5*) and *ALCOHOL DEHYDROGENESE1* (*ADH1*) to give a better understanding of the transcriptional response in both tomato and bittersweet.

In bittersweet, ethylene typically led to ethylene signalling and subsequent up-regulation of *PGB1*, *PGB3*, *RAP2.3* and *RAP2.12* transcripts in both root and shoot tissues (Figure 4). Moreover, the results revealed that ethylene strongly augmented the transcriptional hypoxia response. In bittersweet shoots, seven out of the eight hypoxia-responsive genes were enhanced by ethylene upon hypoxia. In the roots, all 10 genes tested were hypoxia responsive and an ethylene pre-treatment enhanced eight of these 10 transcripts (Figure 4). In tomato root and shoot tissues, ethylene also switched on ethylene signalling and the subsequent up-regulation of *PGB1*, *RAP2.3* and *RAP2.12* transcripts (Figure 5). However, ethylene rarely further increased mRNA levels of hypoxia-responsive genes. In tomato shoot tissues, ethylene only enhanced one (*ACO1*) out of the eight hypoxia-responsive genes during hypoxia. Out of the seven hypoxia-responsive genes in tomato roots, ethylene promoted transcript levels of *ACO1*, *PGB1* and *PDC1* upon hypoxia. Collectively, these results reveal that ethylene enhances hypoxia responses and tolerance in bittersweet, but not in tomato.

## 3. Discussion

In this study, we made a first step in translating the mechanism of ethylene-mediated hypoxia tolerance as established in Arabidopsis [10], to vegetable crops. We show that ethylene can enhance hypoxia tolerance in bittersweet and eggplant, but not in tomato and *S. pennelli*. Moreover, our results reveal that ethylene generally leads to up-regulation of ethylene signaling, *PGB1* and *ERFVII* transcripts in potato, bittersweet and tomato, but that the more downstream hypoxia-responsive genes are not consistently enhanced by early ethylene signaling when O_2_ levels decline (overview in Figure 6). In addition, the capacity to augment the transcriptional hypoxia response through ethylene signaling correlated with higher waterlogging or ethylene-enhanced hypoxia tolerance in the tested *Solanum* species (Figure 6).

While ethylene enhanced transcript levels of *PGB1* and *RAP2.12* orthologues, it is unclear whether this also led to increased corresponding protein levels and biological function. However, our results suggest that ethylene may indeed stimulate ERFVII protein levels in bittersweet and the more waterlogging tolerant potato cultivar Festien, based on the ethylene-enhanced upregulation of typical ERFVII-target hypoxia gene transcripts during subsequent hypoxia treatments (Figure 2 and Figure 4). One explanation could be that, similar to Arabidopsis, ethylene impairs NO-dependent ERFVII proteolysis in these tolerant *Solanum* species through the up-regulation of NO scavenging PGBs, causing enhanced ERFVII stability [10]. Consistently, the lack of *PGB1* up-regulation in the roots of potato cultivar Seresta may explain why ethylene failed to enhance hypoxia-responsive genes in these tissues (Figure 2). Additionally, ethylene may also directly promote protein synthesis of one or more *Solanum* ERFVIIs that escape PRT6 N-degron mediated proteolysis through structural differences, as shown for the ethylene-mediated rice ERFVII SUB1A-1 [51]. Further research is required to determine if ethylene indeed leads to higher *Solanum* ERFVII levels under normoxia, and whether this is mediated through enhanced PGB1-dependent NO scavenging.

Interestingly, while an ethylene pre-treatment increased upstream marker genes *PGB1*, *RAP2.3* and *RAP2.12* transcripts in tomato, it did not enhance mRNA levels of most hypoxia-responsive gene orthologues upon hypoxia (Figure 5 and Figure 6). These results suggest that ethylene-mediated ERFVII levels could in some cases be uncoupled from enhanced target-gene expression upon hypoxia. Strikingly, ethylene did augment three hypoxia-responsive gene transcripts in tomato root tissues, hinting that not all hypoxia-responsive genes are uncoupled from ethylene-enhanced ERFVII regulation in the same fashion. One explanation could be that ethylene only enhances a subset of ERFVII proteins, and that specific ERFVIIs mediate different target genes. Alternatively, it is possible that ethylene does not lead to sufficiently enriched PGB1 protein levels, that PGB1 does not effectively scavenge NO, or that the proteolysis of some ERFVIIs is not NO-dependent in tomato plants and in the roots of the potato cultivar Seresta.

It is currently unclear which hypoxia-responsive genes actually contribute to enhanced hypoxia tolerance. For instance, several studies suggest transcriptional *SUS1* or *ADH1* induction may not be required for anaerobic fermentation, maintenance of glycolysis and subsequent hypoxia tolerance in potato tubers and Arabidopsis, respectively [40,52]. Previous research showed that protein levels of ADH1 in tomato roots increased only 1.7-fold after 24 h of waterlogging, whereas PDC1 levels were even down-regulated [53]. It is likely that the functional implications of specific up-regulated hypoxia genes for hypoxia tolerance are dependent on the plant tissue and determined by the conditions and severity of the stress [6]. In the future, other (ERFVII-dependent) hypoxia-responsive genes should be tested to uncover their role in ethylene-mediated hypoxia acclimation. Especially oxidative stress amelioration is thought to be a major determinant of flooding and hypoxia tolerance [11,27,29,54], and it would therefore be interesting to assess the effect of ethylene on ROS removal during hypoxia and re-oxygenation in *Solanum* species in the future.

Collectively, we show that ethylene-mediated hypoxia responses are conserved in *S. dulcamara* and correspond to enhanced tolerance similar to Arabidopsis and *R. palustris* [9,10]. In potato cultivars, the induction of this mechanism was also conserved in shoots and roots of the waterlogging-tolerant cultivar Festien. Conversely, early ethylene signaling was uncoupled from *PGB1* induction and an enhanced hypoxia response in the roots of waterlogging-intolerant cultivar Seresta, suggesting that waterlogging tolerance in potato could be dependent on ethylene-mediated hypoxia responses in the roots. Similarly, ethylene signaling did not augment most of the transcriptional hypoxia response and hypoxia survival in tomato. While it is unclear whether the (re-)introduction of ethylene-mediated hypoxia genes would enhance flooding tolerance of tomato and more waterlogging sensitive potato cultivars, future work could aim to introduce ethylene-regulated promoter elements into key genes involved in ethylene-mediated hypoxia tolerance (Figure 6). However, it remains to be established why tomato plants and roots of the potato cultivar Seresta do not integrate ethylene signaling to enhance its transcriptional hypoxia response. We conclude that studying genes indicative of ethylene-mediated hypoxia acclimation is a useful approach to explore the universality of this mechanism across species and helps to identify key signaling cascades that confer flooding and hypoxia tolerance in vegetable crops.

## 4. Materials and Methods

### 4.1. Plant Material and Growth Conditions

Plant material: potato (*S. tuberosum* L.) tubers of the elite cultivars Russet Burbank, Kennebec, Festien, Seresta, Avarna and Ambition were obtained from Jan De Haas of HZPC Research B.V, Metslawier, The Netherlands and Mariëlle Muskens of Agrico Research B.V., Bant, The Netherlands. Seeds of bittersweet (*S. dulcamara*) and *S. pennellii* were obtained from Dr. Eric Visser and Dr. Ivo Rieu, respectively, Radboud University, Nijmegen, The Netherlands. Tomato (*S. lycopersicum* L.) of the variety Moneymaker and eggplant (*S. melongena*) seeds were obtained from a local garden center (Intratuin).

Growth conditions potted plants: potato tubers were kept in the dark at room temperature for 1 month, until the eye buds started to sprout. Tubers were placed individually in 11 cm × 11 cm sized square pots filled with a sand soil mixture (1:2), were covered by ~5 cm of soil mixture and stratified at 4 °C in the dark for 3 days. Seedlings of tomato and bittersweet were germinated on agar plates as described below and subsequently transplanted in small round pots (diameter = 5 cm) filled with a sand soil mixture (1:2). Pots were then transferred to a growth chamber for germination/growth under short day conditions (9:00–17:00, temperature (T) = 20 °C, photosynthetic photon flux density (PPFD) = ~130 μmol m^−2^ s^−1^, relative humidity (RH) = 70%). Potato plants were used for waterlogging and ethylene/hypoxia gassing experiments 3 weeks after potting. Tomato and bittersweet plants were 2 weeks old for the shoot survival experiments described below. Per species, homogeneous groups of plants were selected and randomized over treatment groups before phenotypic and molecular analysis. Tomato and bittersweet used for hypoxia survival experiments were transferred back to the same growth room conditions after treatments to recover for 7 days before scoring shoot survival and fresh weight.

Growth conditions seedlings: tomato, bittersweet, eggplant and *S. pennellii* seeds were vapor sterilized through incubation with a beaker containing a mixture of 50 mL bleach (5%) and 3 mL of 37% fuming HCl in a gas tight desiccator jar for 4 h. Seeds were then individually transplanted in 2 rows of 10 seeds on sterile square petri dishes containing 25 mL autoclaved and solidified ¼ MS, 1% plant agar without sucrose supplement. Petri dishes were sealed with gas-permeable tape (Leukopor, Duchefa) and kept at 4 °C in the dark for 3 days. Seedlings were when grown vertically on the agar plates under short day conditions (9:00–17:00, T = 20 °C, PPFD = ~130 μmol m^−2^ s^−1^, RH = 70%) for 7 days, before performing root tip survival experiments and shoot and root harvests for RT-qPCR.

### 4.2. Tolerance Assays, Treatments and Sample Harvests

Waterlogging tolerance of potato cultivars: plants were waterlogged by submerging the pots in large tubs under long day conditions (7:00–23:00, T = 20 °C, PPFD = ~150 μmol m^−2^ s^−1^) in water that was left stagnant 1 day prior to the experiment. During the experiments, the water surface was maintained 1 cm above the soil. Control plants were placed in similar tubs without water. Of each cultivar, 10 pots were taken out of control and waterlogging treatments after 3, 6, 9 and 12 days. Shoot biomass (dry weight) and total leaf area were measured at the start of the treatment (t = 0) and at each subsequent harvest time-point. Root tip hypoxia survival performance assays of tomato, bittersweet, eggplant and *S. pennellii* 7-day old seedlings grown vertically on agar plates were performed as in [10]. Shoot meristem survival was performed on 2-week-old plants grown in pots, as described for Arabidopsis rosettes in [10]. Samples for mRNA analysis of potato, tomato and bittersweet were harvested after 4 h of air and ethylene (~5 μL L^−1^) treatments and after 4 h of subsequent hypoxia (<0.00% O_2_) treatment, similar to [10]. For shoot tissues, 1–4 primary shoot meristems were harvested including the first young leaf. For root tissues, multiple segments (5–10) of ~1 cm long root tips were harvested per sample.

### 4.3. RNA Extraction, cDNA Synthesis and RT-qPCR

RNA extraction, cDNA synthesis, RT-qPCR were performed as described in [10]. Gene orthologue sequences were obtained by BLASTing the protein coding sequences of Arabidopsis genes against the bittersweet transcriptome [45], tomato genome (version SL3.0 and Annotation ITAG3.10) and potato genome [55] using the tblastx tool on [56]. For potato, the orthologue of *ACTIN11* was used as a reference gene. For tomato and bittersweet, the orthologues of *TUBULIN6* and *TIP41* were used as reference genes and the mean CT was used to calculate transcript fold changes. Gene annotations and primers used in this study are listed in Appendix A.

### 4.4. Statistics

The data were plotted and the figures were designed using Graphpad Prism software (San Diego, CA, USA). The statistical tests were performed using either Graphpad Prism (San Diego, CA, USA) or R software and the “LSmeans and “multmultcompView” packages. Potato tolerance data were analyzed through a generalized linear modeling (GLM) approach. A negative binomial error structure was used for the GLM. The other data were analyzed with either a Students *t*-test, 1-way or 2-way ANOVA. If necessary, data was log transformed to meet ANOVA prerequisites. Tukey’s honestly significant difference (HSD) tests were used to correct for multiple comparisons.

## Figures and Tables

**Figure 1 plants-09-01022-f001:**
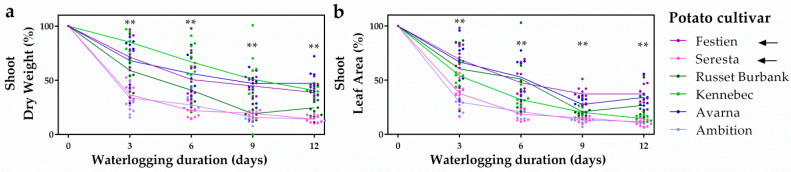
Potato cultivars show variation in waterlogging tolerance. Relative shoot dry weight (**a**), relative leaf area (**b**) of 6 elite potato (*S. tuberosum*) cultivars during 12 days of waterlogging. Values are relative to non-waterlogged plants who were set at 100% per time-point (**a**,**b**). Asterisks indicate significant differences between the Festien (purple) and Seresta (pink) cultivars per time-point and these were the cultivars selected for further analysis (indicated by arrows, ** *p* < 0.01, Generalized linear model, Tukey’s honestly significant difference (HSD), *n* = 10 plants).

**Figure 2 plants-09-01022-f002:**
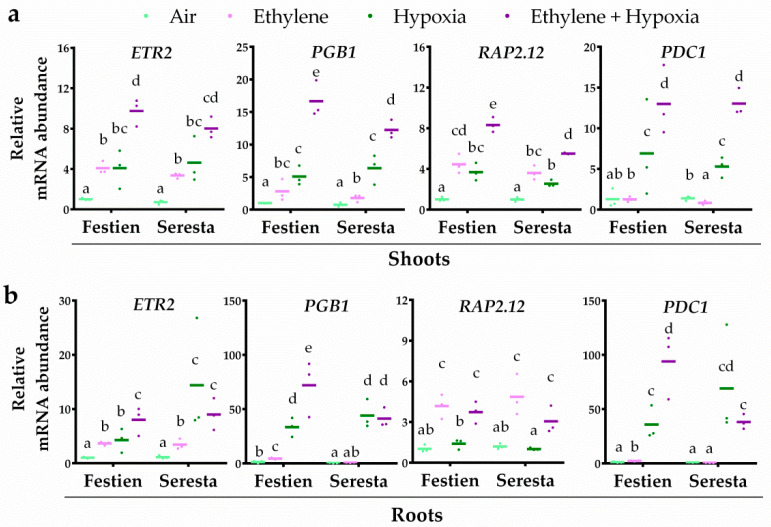
Ethylene differentially modulates hypoxia-responsive transcripts in the roots of elite potato cultivars. Relative mRNA transcript abundance of four marker genes for ethylene-mediated hypoxia tolerance in shoot (**a**) and root tissues (**b**) of the potato cultivars Festien (waterlogging tolerant) and Seresta (waterlogging sensitive) after 4 h of pre-treatment with air (light green) or ~5 μL L^−1^ ethylene (pink), followed by (4 h) hypoxia (green and purple). Marker genes are orthologues of the Arabidopsis genes: ethylene signaling gene *ETR2*, NO-scavenging phytoglobin *PGB1*, ERFVII transcription factor *RAP2.12* and hypoxia adaptive gene *PDC1*. Values are relative to air treated samples of Festien. Different letters indicate significant differences (*p* < 0.05, 2-way ANOVA, Tukey’s HSD, *n* = 3 containing 2 shoot meristems (**a**) or ~10 root tips (**b**)).

**Figure 3 plants-09-01022-f003:**
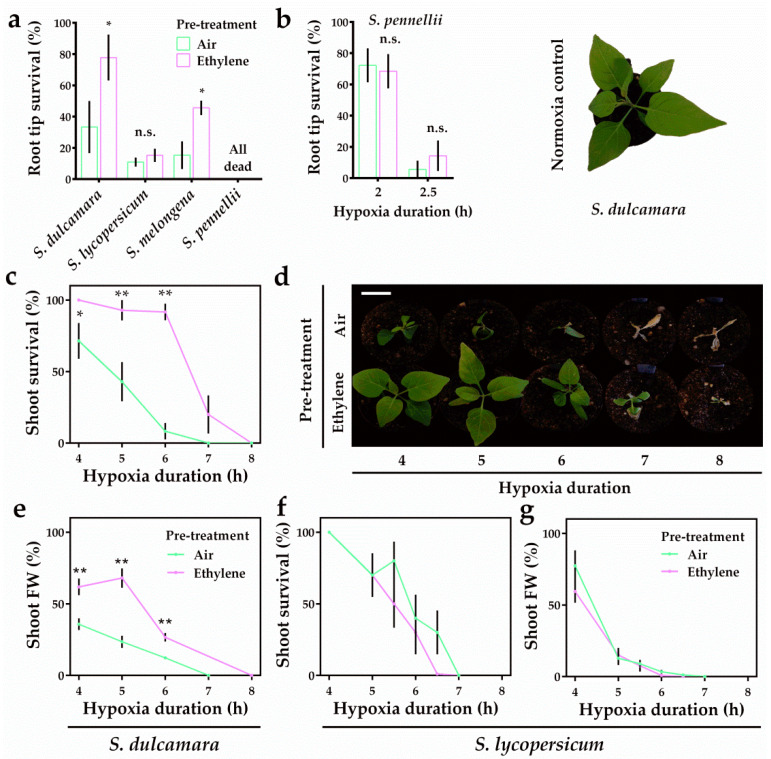
Ethylene differentially mediates hypoxia tolerance in *Solanum* species. Root tip survival of *S. dulcamara*, *S. lycopersicum* (Moneymaker), *S. melongena* and *S. pennellii* seedlings after 4 h of pre-treatment with air (green) or ~5 μL L^−1^ ethylene (purple) followed by 4 h (**a**) or 2 and 2.5 h (**b**) (only *S. pennellii*) of hypoxia and 2 days of recovery. Asterisks indicate significant differences between air and ethylene (error bars are SEM, * *p* < 0.05, Student’s *t* test, *n* = 4–10 rows containing 8–10 seedlings). Shoot survival (**c**,**f**), phenotypes (**d**) (only *S. dulcamara*) and fresh weight (FW) (**e**,**g**) of *S. dulcamara* (**c**–**e**) and *S. lycopersicum* (**f**,**g**) plants after 4 h of air (green) or ~5 μL L^−1^ ethylene (purple) pre-treatment followed by hypoxia and 7 days recovery. Values are relative to control (normoxia) plants. Scale bar = 3 cm. Asterisks indicate significant differences between air and ethylene (error bars are SEM, ** *p* < 0.01, Student’s *t* test, *n* = 10–13 plants).

**Figure 4 plants-09-01022-f004:**
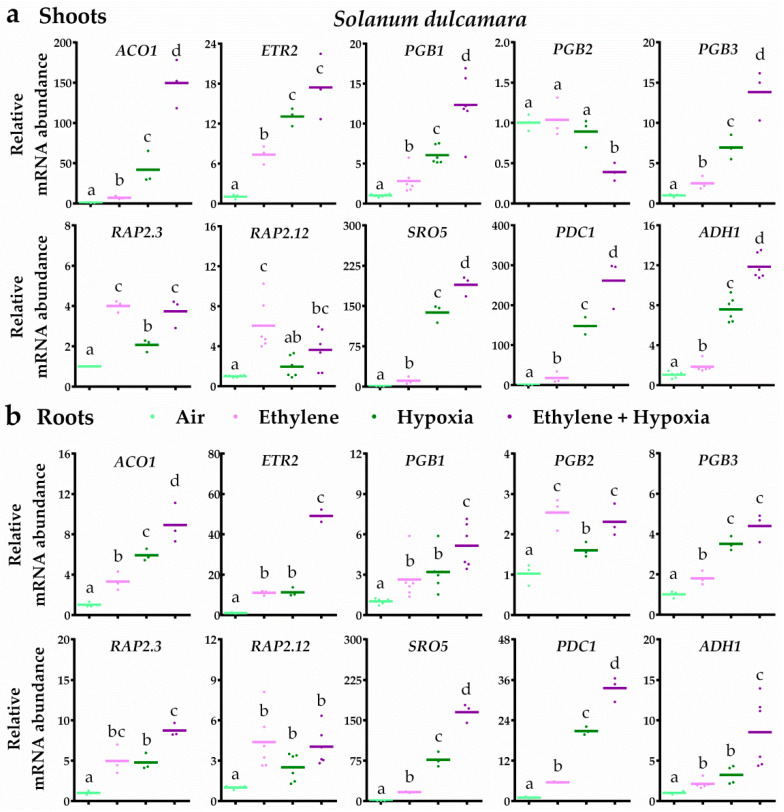
Ethylene augments hypoxia-responsive transcripts in bittersweet. Relative mRNA transcript abundance of 10 marker genes for ethylene-mediated hypoxia tolerance in bittersweet (*S. dulcamara*) shoot (**a**) and root tissues (**b**) after 4 h of pre-treatment with air (light green) or ~5 μL L^−1^ ethylene (pink), followed by (4 h) hypoxia (green and purple). Marker genes are orthologues of the Arabidopsis genes: ethylene signaling genes *ACO1* and *ETR2*, NO-scavenging phytoglobins *PGB1*, *PGB2* and *PGB3*, ERFVII transcription factors *RAP2.3* and *RAP2.12* and hypoxia adaptive genes *SRO5*, *PDC1* and *ADH1*. Values are relative to air treated samples. Different letters indicate significant differences (*p* < 0.05, 1-way ANOVA, Tukey’s HSD, *n* = three or six biological replicates containing 2–4 shoot meristems (**a**) or ~10 root tips (**b**)).

**Figure 5 plants-09-01022-f005:**
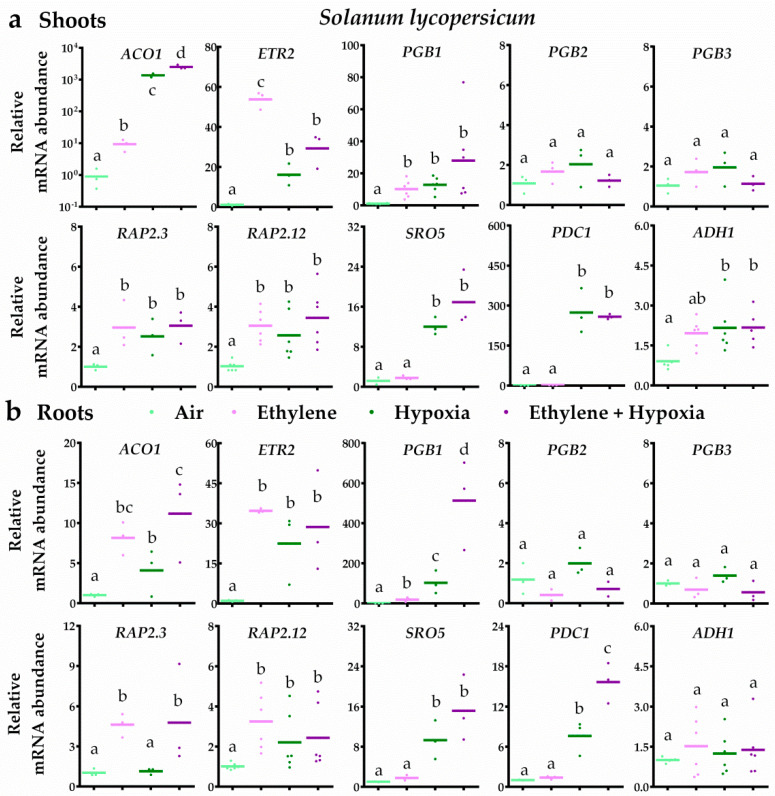
Ethylene rarely enhances hypoxia-responsive transcripts in tomato. Relative mRNA transcript abundance of 10 marker genes for ethylene-mediated hypoxia tolerance in tomato (*S. lycopersicum* cv MoneyMaker) shoot (**a**) and root tissues (**b**) after 4 h of pre-treatment with air (light green) or ~5 μL L^−1^ ethylene (pink), followed by (4 h) hypoxia (green and purple). Marker genes are orthologues of the Arabidopsis genes: ethylene signaling genes *ACO1* and *ETR2*, NO-scavenging phytoglobins *PGB1*, *PGB2* and *PGB3*, ERFVII transcription factors *RAP2.3* and *RAP2.12* and hypoxia adaptive genes *SRO5*, *PDC1* and *ADH1*. Values are relative to air treated samples. Different letters indicate significant differences (*p* < 0.05, 1-way ANOVA, Tukey’s HSD, *n* = three or six biological replicates containing 2–4 shoot meristems (**a**) or ~10 root tips (**b**)).

**Figure 6 plants-09-01022-f006:**
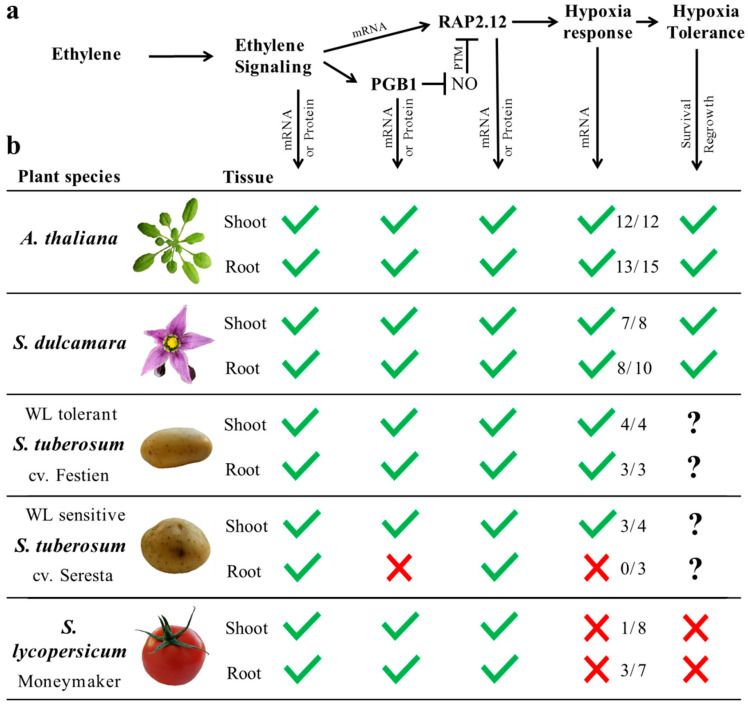
Overview of ethylene-mediated hypoxia responses in *Solanum* species. (**a**) Schematic showing the proposed mechanism of ethylene-mediated hypoxia tolerance, as discovered in Arabidopsis [10]. Arrows pointing downward indicate key processes that were verified in Arabidopsis at the mRNA and protein level, or tested in the *Solanum* species at mRNA level in both roots and shoots. (**b**) Table showing whether marker genes or hypoxia tolerance (as assessed by regrowth capacity following stress removal) were enhanced by an early ethylene treatment (green check mark = yes, red cross = no, question mark = not tested). The n/n in the hypoxia response column indicates the amount of hypoxia genes that are ethylene-enhanced and hypoxia-responsive upon hypoxia compared to the total amount of hypoxia-responsive genes tested. This table is based on the experimental data shown in Figure 1, Figure 2, Figure 3, Figure 4 and Figure 5 and [10,27].

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
