# Peer review of "Ethylene Differentially Modulates Hypoxia Responses and Tolerance across Solanum Species"

_plants, 2020, doi:10.3390/plants9081022_

Round 1
Reviewer 1 Report
The manuscript entitled " Ethylene differentially modulates hypoxia responses and tolerance across Solanum species" deals with the issue how cultivated and wild Solanum species integrate ethylene signaling to control subsequent hypoxia acclimation. The authors have studied the transcript levels of selected marker genes whose upregulation is indicative of ethylene-mediated hypoxia acclimation in Arabidopsis.
Introduction is written clearly and correctly, constituting an interesting introduction to this topic. I suggest that the authors provide some relevant information as to why the study of flooding hypoxia is so important in the context of the Solanaceae plant family (increased food demand, global warming etc.) Please provide some economic data emphasizing the importance of such experiments. There should also be more information in the manuscript about the role of ERF-VII family factors in plants.
Materials and methods are properly selected and the conclusions drawn from the results are correct.
The discussion is very structured and logical.
Minor comments:
Line 298 No information about the concentration of sodium hypochlorite in bleach
Author Response
Reviewer 1:
We thank the reviewer for his or her time and positive feedback on the manuscript. We have processed the comments point by point below.
The manuscript entitled " Ethylene differentially modulates hypoxia responses and tolerance across Solanum species" deals with the issue how cultivated and wild Solanum species integrate ethylene signaling to control subsequent hypoxia acclimation. The authors have studied the transcript levels of selected marker genes whose upregulation is indicative of ethylene-mediated hypoxia acclimation in Arabidopsis.
Introduction is written clearly and correctly, constituting an interesting introduction to this topic. I suggest that the authors provide some relevant information as to why the study of flooding hypoxia is so important in the context of the Solanaceae plant family (increased food demand, global warming etc.) Please provide some economic data emphasizing the importance of such experiments. There should also be more information in the manuscript about the role of ERF-VII family factors in plants.
Additional information is added with regards to global warming and food demand (line 30-34), the role of ERFVIIs and the N-degron pathway for plant development and stress acclimation (line 49-50) and the economical value of tomato (line 76-77).
Materials and methods are properly selected and the conclusions drawn from the results are correct.
The discussion is very structured and logical.
Minor comments:
Line 298 No information about the concentration of sodium hypochlorite in bleach
The concentration of bleach is added now (5%, line 321).
Reviewer 2 Report
This manuscript is a clear continuation of the group's topic on ethylene-mediated adaptation to hypoxia. This time, on how different cultivated and wild Solanum species integrate flooding cues and ethylene to regulate hypoxia adaptation.
1) In figure 1 and corresponding result section.
Do ethylene treatments have any effect on the response to waterlogging of these potato cultivars?
If you have the data, it would be great to have a panel in Figure 2 maybe with the two selected cultivars.
2) First paragraph of result sections 2.2 and 2.4.
Prepare the reader for what to expect. Integrate this information with what is already know (you recently published a great article). Step by step guide the reader. Why is transcription key here, why not protein stability? Prepare the stage for your discussion. Elaborate more your results, digest them a bit more for the reader. Why did you chose these marker genes for potato and why do you have a larger group of markers for tomato?
3) Lines 126-127. Together these data suggest that the capacity to enhance hypoxia response genes through ethylene signaling correlates with waterlogging tolerance in these two potato cultivars.
"Correlates" or it is in agreement?
As pointed out earlier in figure 1, we do not have any information on how ethylene might be affecting the response to hypoxia at the physiological level.
Could you please elaborate a bit more on this sentence? This sentence is difficult to understand. Digest it a bit for the reader.
I might be missing something but the four chosen marker genes in shoots behave very similarly between the two cultivars, same trends. In roots, there is some similarities too. The only big difference is seen on ET+Hypoxia treatment for PGB1 and PDC1. Could you explain a bit more? I do not see the correlation. Prepare me for figure 6, I love figure 6 but you need to show me where these differences are. It is not clear to me at this point.
Initial levels might also play a key role in the response. I am assuming the value 1 for Festien and Seresta for all the markers is totally arbitrary. If Air Festien is compared to Air Seresta, do you find large differences? This might be important.
4) First paragraph of result sections 2.3.
Transition? After the potato results, it feels as if the manuscript started again here but now talking about tomato.
Why are you testing or quantifying new phenotypes that were not taken into account for potato? Any specific reason?
5) Figure 3d
Would it be possible to add time 0 h? To have the normoxia control.
6) Paragraph from line 178 - 189.
Elaborate a bit more and prepare the reader for what you are going to say in the discussion. Why are the highlighted results important? Do the results match your initial hypothesis?
Which specific markers are potentially key for the physiological differences observed between bittersweet and tomato based on your results? You can explain why in the discussion but at this point the reader needs a bit of help.
7) Line 188-189. Collectively, these results reveal that ethylene strongly enhances hypoxia responses and tolerance in bittersweet, but not in tomato.
Very strong statement. Would you rather say: "these results suggest that ethylene upregulates hypoxia gene markers that can strongly contribute to the response and tolerance to O2 low levels in bittersweet, whereas it is not the case in tomato"? Or something similar. Not trying to re-write your text, we just need to be careful with how rigorous our statements are.
8) Figure 6.
Excellent summary for the manuscript. Work on the description of the results shown in figure 2 and 5 to make this figure shine. As stated above, it is not clear to me how large the differences in expression of markers are between cultivars.
Line 260. Conversely, early ethylene signaling was uncoupled from an enhanced hypoxia response in the roots of waterlogging-intolerant cultivar Seresta, suggesting that waterlogging tolerance in potato could be dependent on ethylene-induced hypoxia responses in the roots.
Dependent or independent?
Author Response
Reviewer 2:
We especially thank the reviewer for his or her time and very valuable and critical feedback on the manuscript. We think the comments have increased the readability and clarity of the manuscript and have highlighted our response and revisions below.
This manuscript is a clear continuation of the group's topic on ethylene-mediated adaptation to hypoxia. This time, on how different cultivated and wild Solanum species integrate flooding cues and ethylene to regulate hypoxia adaptation.
1) In figure 1 and corresponding result section.
Do ethylene treatments have any effect on the response to waterlogging of these potato cultivars?
If you have the data, it would be great to have a panel in Figure 2 maybe with the two selected cultivars.
Unfortunately, we do not have such data as it is known that ethylene passively accumulates in flooded/waterlogged tissues (added to line 107) to potentially mediate adaptive responses, and we therefore think that additional ethylene application would not be of further benefit to the plant (as ethylene signaling would naturally be saturated). The question we try to answer is how ethylene is used to mediate the transcriptional hypoxia response. We therefore aim to untangle the integration of this ethylene signal with the hypoxia response of the 2 cultivars in Figure 2, and observe whether differences corresponds to waterlogging tolerance. In the future, it would be interesting to repeat this experiment with chemical inhibition of ethylene signaling (1-MCP), to uncover the functional role of ethylene for waterlogging acclimation/tolerance in potato.
2) First paragraph of result sections 2.2 and 2.4.
Prepare the reader for what to expect. Integrate this information with what is already know (you recently published a great article). Step by step guide the reader. Why is transcription key here, why not protein stability? Prepare the stage for your discussion. Elaborate more your results, digest them a bit more for the reader. Why did you chose these marker genes for potato and why do you have a larger group of markers for tomato?
We have now introduced the rationale of these result sections more thoroughly as suggested, using these comments (line 119-120, 129-130 and 187-188). We believe the rationale for selecting these genes is now more clear from these paragraphs.
3) Lines 126-127. Together these data suggest that the capacity to enhance hypoxia response genes through ethylene signaling correlates with waterlogging tolerance in these two potato cultivars.
"Correlates" or it is in agreement?
We have adjusted this phrase to “corresponds to”, added the word : “root”, and highlight this could in this case be dependent on PGB1 induction to prepare the reader for the discussion (line 143-144.
As pointed out earlier in figure 1, we do not have any information on how ethylene might be affecting the response to hypoxia at the physiological level. Could you please elaborate a bit more on this sentence? This sentence is difficult to understand. Digest it a bit for the reader. I might be missing something but the four chosen marker genes in shoots behave very similarly between the two cultivars, same trends. In roots, there is some similarities too. The only big difference is seen on ET+Hypoxia treatment for PGB1 and PDC1. Could you explain a bit more? I do not see the correlation. Prepare me for figure 6, I love figure 6 but you need to show me where these differences are. It is not clear to me at this point. Initial levels might also play a key role in the response. I am assuming the value 1 for Festien and Seresta for all the markers is totally arbitrary. If Air Festien is compared to Air Seresta, do you find large differences? This might be important.
We have elaborated this more in the text (throughout paragraph 2.2). Our rationale is that, if plants integrate ethylene signaling to augment their (transcriptional) hypoxia reponse, this could contribute to fast acclimation and higher tolerance. In the roots of the more intolerant cultivar Seresta, ethylene does not bolster the hypoxia responsiveness of ETR2, PGB1 and PDC1. Similarly, waterlogging tolerance (where only the roots would experience hypoxia stress) was reduced in this cultivar. No clear differences were observed in initial levels of the transcripts between cultivars, as all mRNA levels were relative to air treated samples of Festien (as written in the legend). We therefore “suggest that the capacity to enhance root hypoxia response genes through ethylene signaling correlates with waterlogging tolerance in these two potato cultivars.” (we added the word root, as this is indeed essential, line 143-144).
4) First paragraph of result sections 2.3.
Transition? After the potato results, it feels as if the manuscript started again here but now talking about tomato.
Why are you testing or quantifying new phenotypes that were not taken into account for potato? Any specific reason?
We agree this transition may seem quite abrupt and have amended the introduction of this paragraph (line 157-159). Technically it was not possible for us to reliably assess potato plant hypoxia tolerance grown from tubers as (combination of) (1) we had limited access to tubers, (2) relatively high variation between tuber grown plants already under control conditions (3), hypoxia chambers are of restricted size to fit enough tuber-grown plants to assess tolerance at sufficient sample size. Therefore, we switched to Solanum species grown from seeds as we (1) had previously established differential tolerance in roots of tomato and bittersweet, (2) had plenty of seeds to optimize hypoxia tolerance experiments (3) and could achieve high enough sample size to reliably test their hypoxia tolerance after ethylene treatment.
5) Figure 3d
Would it be possible to add time 0 h? To have the normoxia control.
We added an image of a normoxia control plant to Figure 3d.
6) Paragraph from line 178 - 189.
Elaborate a bit more and prepare the reader for what you are going to say in the discussion. Why are the highlighted results important? Do the results match your initial hypothesis?
Which specific markers are potentially key for the physiological differences observed between bittersweet and tomato based on your results? You can explain why in the discussion but at this point the reader needs a bit of help.
We have added some additional rationale to this paragraph (line 187-188), but feel that too much detail and speculation here will remove focus and is better fitted for the discussion. We touch upon this in the fourth paragraph of the discussion.
7) Line 188-189. Collectively, these results reveal that ethylene strongly enhances hypoxia responses and tolerance in bittersweet, but not in tomato.
Very strong statement. Would you rather say: "these results suggest that ethylene upregulates hypoxia gene markers that can strongly contribute to the response and tolerance to O2 low levels in bittersweet, whereas it is not the case in tomato"? Or something similar. Not trying to re-write your text, we just need to be careful with how rigorous our statements are.
We agree that the statement was too strong and have amended it accordingly.
8) Figure 6.
Excellent summary for the manuscript. Work on the description of the results shown in figure 2 and 5 to make this figure shine. As stated above, it is not clear to me how large the differences in expression of markers are between cultivars.
Thank you, we have also added photographic images of the various species to make the summarizing figure more appealing and intuitive to non-specialist readers.
Line 260. Conversely, early ethylene signaling was uncoupled from an enhanced hypoxia response in the roots of waterlogging-intolerant cultivar Seresta, suggesting that waterlogging tolerance in potato could be dependent on ethylene-induced hypoxia responses in the roots.
Dependent or independent?
Dependent. In the tolerant cultivar Festien, the transcriptional root hypoxia response was modulated by ethylene. In the intolerant cultivar Seresta this is not the case and this corresponds to reduced waterlogging (root flooding) tolerance, suggesting that tolerance could potentially be driven by ethylene modulation of hypoxia responses (through PGB1). We have added: “PGB1 induction and” to the sentence to emphasize why it could be dependent on this mechanism (line 284).